# Fungal Aeroallergen Sensitization Patterns among Airway-Allergic Patients in Zagazig, Egypt

**DOI:** 10.3390/jof9020185

**Published:** 2023-01-30

**Authors:** Ghada A. Mokhtar, Manar G. Gebriel, Noha M. Hammad, Sylvia W. Roman, Osama Attia, Ahmed Behiry, Nagwan A. Ismail, Mohamed Salah Abd El Azeem El Sayed, Ahmed Nagy Hadhoud, Yosra A. Osama, Ahmed A. Ali, Heba M. Kadry

**Affiliations:** 1Department of Medical Microbiology and Immunology, Faculty of Medicine, Zagazig University, Zagazig 44519, Egypt; 2Department of Internal Medicine, Allergy and Clinical Immunology, Faculty of Medicine, Ain Shams University, Cairo 12613, Egypt; 3Department of Internal Medicine, Faculty of Medicine, Zagazig University, Zagazig 44519, Egypt; 4Department of Tropical Medicine and Endemic Diseases, Faculty of Medicine, Zagazig University, Zagazig 44519, Egypt; 5Department of Chest Diseases, Faculty of Medicine, Zagazig University, Zagazig 44519, Egypt; 6Department of Otorhinolaryngology, Faculty of Medicine, Zagazig University, Zagazig 44519, Egypt; 7Department of Clinical Pathology, Faculty of Medicine, Zagazig University, Zagazig 44519, Egypt; 8Department of Pediatrics, Faculty of Medicine, Zagazig University, Zagazig 44519, Egypt

**Keywords:** allergic-airway diseases, *Alternaria alternata*, Egypt, fungal sensitization, mixed mold, skin prick test, specific serum IgE, Zagazig

## Abstract

Background: Airway allergies such as asthma and allergic rhinitis, as well as their comorbidities, are increasing worldwide, causing significant socioeconomic health burdens to societies. It is estimated that between 3% and 10% of the population is allergic to fungi. The type of fungal sensitization varies from one geographical region to another. The present study aimed to identify the common fungal aeroallergen sensitization patterns among airway-allergic patients residing in the Zagazig locality, Egypt, in order to obtain a better understanding of fungal allergy, in addition to improving the awareness and management strategies for those patients. Methods: The present cross-sectional study included 200 allergic rhinitis and asthma patients. Sensitization to fungal aeroallergens was evaluated by skin prick testing and in vitro measurement of total and specific immunoglobulin E. Results. As determined by a skin prick test, 58% of the patients studied were allergic to mixed molds. *Alternaria alternata* was the predominant fungal aeroallergen among the studied patients (72.2%), which was followed by *Aspergillus fumigatus* (53.45%), *Penicillium notatum* (52.6%), *Candida albicans* (34.5%), and *Aspergillus niger* (25%). Conclusion: Mixed mold sensitization ranked fourth among the most frequent aeroallergens in airway-allergic patients, and *Alternaria alternata* was the most frequently encountered fungal aeroallergen in the Zagazig locality.

## 1. Introduction

Atopic diseases (immunoglobulin E (IgE)-mediated allergic diseases), such as allergic rhinitis, asthma, and atopic dermatitis, have been increasing in recent years. Collectively, they affect approximately 20% of the population worldwide [1]. Allergens in indoor and outdoor environments can trigger or aggravate allergic diseases. Allergens vary from region to region; some could be autochthonous to a particular geographical area. Therefore, to diagnose and treat atopic patients, it is essential to inspect the local environment to identify the most predominant allergens [2].

According to past reports from Egypt and Saudi Arabia, *Aspergillus, Alternaria*, *Cladosporium*, *Penicillium, Ulocladium, Fusarium,* and yeast spores were the most frequently air-suspended fungal spores [3,4,5]. Moreover, it was well established that the distribution of air-suspended fungal spores in open sites of any country is nearly identical, with only minor quantitative differences [6]. In Egypt, fungal air spore concentrations indoors and outdoors were higher than reported from other parts of the world. Furthermore, Egyptian homes’ indoor air quality may be poor since spore concentration surpassed the 500 CFU/m^2^ limit proposed by the World Health Organization (WHO) [7,8]. Temperature, rain, and relative humidity are the factors that influence fungal spore suspension in air. While high temperatures enhance spore production, heavy rains wash off spores and considerably decrease their air concentrations and exposure risk. Additionally, relative humidity reduces dry air spore concentrations such as *Cladosporium* and *Alternaria* [9]. 

Although the precise prevalence of sensitization to fungi is unknown, it is estimated to range between 3% and 10% among the general population (depending extensively on the weather conditions of the area under study) [10] and can reach 50% in the inner cities [11]. According to reports, the most common fungal aeroallergens are *Penicillium*, *Aspergillus*, *Cladosporium*, and *Alternaria* [12]. The spores of the first two genera are present year-round [10], whereas *Cladosporium* and *Alternaria* have seasonal maxima [13].

Fungal spores are extensively air-suspended indoors and outdoors. Among the over 100,000 reported fungal species, a few hundred have been identified as opportunistic pathogens [14,15]. Exposure to fungal spores induces pathology in the human body via three distinct mechanisms: direct infection, induction of immune dysregulation, and toxic effect of fungal secondary metabolites [16]. Moreover, combined allergic and infective stimuli result in several clinical presentations characterized by evidence of lung damage [17].

Poor asthma control is correlated with fungal sensitization, resulting in recurrent asthma attacks [18], an increased need for corticosteroids, an increase in hospital admissions, and asthma-related mortalities [19,20]. In addition, sensitization to fungal allergens jeopardizes the lungs of allergic airway patients to progressive damage [21]. Fungal airway colonization appears to be the potential trigger for allergic fungal airway disease, but the place of antifungals in patient management remains doubtful [21]. However, it has been proven that severe asthmatic patients sensitized to fungal allergens may benefit from fungal immunotherapy [22]. In practice, fungal aeroallergen sensitization patterns can be identified by a skin prick test (SPT) or fungal-specific IgE immunoassays such as immunoblot [23].

Since the concentration of fungal spores in the atmosphere surpasses 1000 times pollen counts [24] and fungal spore season persists twice the duration of the pollen season, fungal-sensitized patients are usually unaware of the source of exposure and can only determine that symptoms are perennial [25,26]. In order to provide accurate data to public health, prevent asthma exacerbations, and adopt better management strategies for allergic patients, the knowledge gap related to fungal aeroallergen sensitization patterns in our locality needs improvement. The study aimed to investigate trends in the prevalence of sensitization to airborne fungal allergens among patients attending the Zagazig Allergy and Immunology Unit (AIU). In addition, it aimed to analyze specific sensitization patterns among the five most essential fungi: *Aspergillus niger (A. niger), Aspergillus fumigatus (A. fumigatus), Alternaria alternate (A. alternate), Penicillium notatum (P. notatum)* and *Candida albicans (C. albicans).*

## 2. Materials and Methods

### 2.1. Patients

The present cross-sectional study included airway allergy patients from attendants to the AIU, Faculty of Medicine, Zagazig University, Zagazig (30°33′59.99″ N and 31°29′59.99″ E), which is located 100 kilometers northeast of Cairo (30°03′17.40″ N and 31°13′26.40″ E) [27], the capital of Egypt. Egypt’s climate is semi-desert, and it is characterized by hot, dry summer (June–August) months, warm autumn (September–November) months, moderate winter with little rainfall (December–February) months, and warm spring (March–May) months [28]. The study was implemented during the period from June 2022 to September 2022. During the study period, the average high temperature was 96° F, and the average low was 74° F [29]. As regards the prevalence of fungal allergens (15.57%) [30] and attendance rate in AIU (1056 airway-allergic patients within six months), the sample size calculated, with a confidence interval of 95% and power of 80%, was 166 patients. Due to the potential non-response rate, the required sample size has increased to 200 patients by allowing an additional 20% in sample recruitment. 

All patients were subjected to complete clinical history, including sociodemographic data. The diagnosis of allergic asthma and allergic rhinitis adhered to the Global Initiative of Asthma (GINA) and Allergic Rhinitis and its Impact on Asthma (ARIA) guidelines, respectively [31,32]. Exclusion criteria for participation in this study were pregnancy and lactation, chronic illnesses, malignancy, autoimmunity, patients who were unable to stop antihistamine one week before skin prick test (SPT), dermographism, severe eczema, history of specific immunotherapy and patients’ refusal.

### 2.2. Allergen Extract Preparation 

Coca’s extracted aeroallergens were prepared and standardized under complete aseptic conditions at the AIU laboratory, as previously described [33,34]. Prepared aeroallergens were labeled and stored at 2–8 °C until use. The allergen panel prepared and employed in SPT included ten common aeroallergens: mixed molds (*Alternaria*, *Aspergillus*, *Penicillium*, and *Cladosporium species*), mixed mites (*Dermatophagoides pteronyssinus* and *Dermatophagoides farinae*), date palm pollen, smoke, hay dust, wool, cotton, mixed feather (pigeon, duck, goose, and chicken) cat hair, and dog hair. 

### 2.3. Skin Prick Test

According to the SPT practical guide of the European Academy of Allergy and Clinical Immunology (EAACI), the SPT was performed, read, and interpreted [35]. Histamine dihydrochloride (10 mg/mL) and glycerinated saline were employed as positive and negative controls, respectively. 

### 2.4. Blood Sampling

For immunoassays, 3 mL of whole peripheral blood was collected. After centrifugation, the serum was stored at −20 °C until processing. 

### 2.5. Testing of Serum Total and Specific IgE 

Total IgE was measured using an indirect enzyme-linked immunosorbent assay according to the manufacturer’s instructions (Chemux Bioscience, South San Francisco, CA, USA). Total IgE serum levels up to 150 IU/mL were considered normal.

The serum aeroallergen-specific IgE for 13 common aeroallergens (*Dermatophagoides farinae, Dermatophagoides pteronyssinus, A. alternata, A. niger, A. fumigatus, P notatum, C. albicans*, mixed grasses, birch, cat epithelium, dog epithelium, feather mixture, and cockroach) was analyzed by immunoblot assay according to the manufacturer company protocol (AllergyScreen^®^ system, MEDIWISS Analytic GmbH, Moers, Germany). Serum-specific IgE was analyzed by a rapid scanner (Improvio Scanner System, Moers, Germany). The test was considered valid if the process control on each strip’s first position was colored. Specific IgE level ≥ 0.35 IU/mL was deemed positive.

### 2.6. Statistical Analysis

Collected data were statistically analyzed using SPSS software (version 24; IBM Corp., Chicago, IL, USA). Quantitative variables were displayed as mean ± standard deviation (SD) and median (min–max). Nominal and ordinal variables were displayed as frequency and percentage. Pearson Chi-Square (χ2) and Fisher’s exact tests were used when appropriate to analyze the association between dependent variables (SPT, total, and specific IgE) and independent variables (age, sex, residence, type of airway allergy, and associated allergic conditions). The Mann–Whitney U test was used to compare the median difference between total IgE serum levels (dependent variable) in the dichotomous variable (independent variable). Spearman’s correlation coefficient (r) was used to determine the degree and direction of association between total IgE serum level (dependent variable) and age (independent variable). *p*-values less than or equal to 0.05 were considered statistically significant.

## 3. Results

### 3.1. Sociodemographic and Clinical Characteristics 

Two hundred patients with allergic airway diseases and attending AIU for skin testing of aeroallergen sensitization were selected by systematic random sampling to participate in the present study. More than two-thirds of the participants (73%) were adults, with males outnumbering females by 60.5%. Approximately half of the participants (58.0%) were rural population. Over half of the participants (60.5%) were diagnosed with allergic rhinitis, while 27% were diagnosed with combined allergic rhinitis and asthma, all significantly correlated with allergic rhinitis (*p* = 0.038). The sociodemographic and clinical characteristics of the studied patients are illustrated in Table 1.

### 3.2. Aeroallergens Sensitization Patterns by SPT

Mixed mold sensitization ranked fourth among the most frequent aeroallergen sensitizations detected by SPT in allergic patients (58%) (Figure 1). It was found that 97.5% of the studied patients demonstrated SPT reactivity to at least one of the ten tested aeroallergens. In mixed mold SPT-positive patients, only one patient (0.9%) was monosensitized to mixed mold allergen. Polysensitization was significantly associated with exposure to two to four aeroallergens simultaneously (*p* = 0.001), and 5.2% of mixed mold SPT-positive patients were polysensitized to more than five aeroallergens other than mixed molds (Figure 2).

### 3.3. Fungal Aeroallergens Sensitization Patterns by Immunoblot Assay

*Alternaria alternata* was the most prevalent fungal aeroallergen in mixed molds sensitized-allergic patients (72.2%), which was followed by *A. fumigatus* (53.45%), *P. notatum* (52.6%), *C. albicans* (34.5%), and *A. niger (25%)*. The results of fungal aeroallergen sensitizations by immunoblot assay were significantly correlated with mixed mold sensitization by SPT (*p* < 0.001) (Figure 3). Additionally, 45.5% of the studied patients were polysensitized to at least two fungal aeroallergens (Figure 4). The monosensitization rate of each fungal allergen in patients sensitized to only one fungal aeroallergen is demonstrated in Figure 5.

### 3.4. Correlations of Sociodemographic and Clinical Characteristics to Total and Specific IgE 

Both the sociodemographic and clinical characteristics of the studied patients did not correlate with different patterns of fungal aeroallergen sensitization (Table 2 and Table 3). In allergic rhinitis, asthma, and combined airway allergy patients, sensitization to mixed molds was prevalent in 57.7%, 48%, and 61%, respectively. The sensitivity, specificity, and diagnostic accuracy of specific IgE immunoblot assay to fungal aeroallergen sensitization was 98.3%, 92.9%, and 96%, respectively (Table 2). However, *C. albicans* sensitization was observed in adults (*p* = 0.04), allergic rhinitis patients (*p* = 0.02), and patients with associated allergic conjunctivitis (*p* = 0.05), with statistical significance. *A. alternata* sensitization was significantly associated with sensitization to *A. fumigatus* (*p* < 0.001), *P. notatum* (*p* < 0.001), *A. niger* (*p* < 0.001), and *C. albicans* (*p* = 0.035) (Table 3).

There was no significant difference in total IgE serum level between sensitized and non-sensitized patients to mixed molds aeroallergen (*p* = 0.7), as tested by SPT (Figure 6). A non-significant correlation was detected between the age of the studied patients and total serum IgE level (*r* = −0.01, *p* = 0.2) (Figure 7). In addition, 44% of the studied patients had normal values of total IgE serum level (up to 150 IU/mL), while 1.5% had values >1000 IU/mL. The correlation between total IgE serum level and sociodemographic and clinical characteristics and fungal sensitization patterns was not statistically significant (data were not presented).

## 4. Discussion

The pathogenesis of atopic diseases depends on aeroallergens, which have recently gained popularity in clinical research worldwide [36]. Although fungi are a rich source of allergenic molecules with a wide array of molecular structures, including cell wall components, enzymes, toxins, and phylogenetically highly conserved cross-reactive proteins [17], fundamental research and the clinical application did not consider this fact [26].

Consistent with previous studies [37,38], SPT revealed that mixed mites extract was the most prevalent aeroallergen (83.5%) in our studied population. Despite Egypt’s dry and moderate climate, the increased use of air conditioners has contributed to an increase in indoor humidity. However, due to the agricultural nature of the area, date palm pollen (74.5%) serves as a secondary allergenic source.

In line with previous reports [25,37,39], our SPT results revealed that sensitization to fungi comes in the fourth rank after mixed mites, pollen, and smoke as an aeroallergen source implicated in airway sensitization in 58% of the allergic patients. Although the precise prevalence of fungal sensitization is unknown, it is speculated to reach 10% in the general population. Nevertheless, it has been previously estimated that allergenic fungi can incite an allergic response in 19–45% of atopic patients during skin testing [9]. Furthermore, in atopic patients, the prevalence of fungal sensitization diverges markedly depending on different demographic, occupational, geographical, environmental, and methodological elements [10,40,41]. 

Except for one patient, all patients sensitized to mixed molds allergen extract in the study (99.1%) displayed sensitization to other allergens, indicating that monosensitization to fungi is unusual [12]. The increasing number of sensitizations presumably characterizes the natural history of atopic patients. It may represent a typical evolution of allergy, which can be caused by numerous factors, including hereditary, environmental factors, and cross-sensitization between allergen-sharing common allergenic epitopes. Moreover, a functional defect of T regulatory cells was another hypothesis for explaining the trend to develop polysensitization [42]. It has been reported that fungi can trigger the immune system and potentiate the inflammatory response elicited by other allergens such as pollen. [43].

Airborne fungi are present in indoor and outdoor environments, sometimes occupying 70% of the total microflora of air [2]. *Aspergillus* spp. and *Penicillium* spp. are among the predominant indoor molds [44], while *Alternaria* spp. and *Cladosporium* spp. are typically outdoors. In Egypt, *Alternaria, Aspergillus, Cladosporium, Penicillium,* and yeasts are the most abundant airborne fungi indoors and outdoors in both urban and rural areas [7]. Although *Alternaria* spp. is air-suspended at lower concentrations than other fungal aeroallergen spores, it demonstrated the highest sensitization rate in atopic patients in the current and previous studies [9,45]. Compared to Fernandez-Soto et al. [18], who estimated that *A. alternata* sensitization ranges from 3.6 to 39.4%, our results revealed that the sensitization rate to *A. alternata* (43%) was the highest among the studied population, which was followed by indoor fungal spores, *A. fumigatus* (32%) and *P. notatum* (30.5%), and *C. albicans* (23%). Both outdoor and indoor fungi demonstrated nearly the same trend of sensitization rates when correlating to the sociodemographic and clinical characteristics of the studied patients. Since sensitization to outdoor fungal spores is more prevalent than indoor fungal spores [46], *A. alternata* sensitization was evident in 72.4% of the SPT fungal-sensitized patients, which was comparable to Mari et al. (66.1%) [12]. Furthermore, in Egypt, the highest peaks of *Alternaria* spp. air-suspended spores were recorded in June, July, and August [47], coinciding with the study period.

SPT is the most reliable and sensitive tool to diagnose IgE-mediated allergic diseases and is considered the gold standard for allergy testing. It is a low-cost, minimally invasive procedure with immediate results that is reproducible when performed by professional allergists [48]. Considering SPT as a gold standard, the specific IgE immunoreactivity to fungal allergens by immunoblot assay exhibited 96% diagnostic accuracy, 98.3% sensitivity, and 96.9% specificity.

Grouping the fungal allergen-sensitized patient by the pattern of specific IgE immunoblot reactivity elucidated demonstrated a prominent trend to combined sensitization to different fungal allergens in 45.5%. In contrast, monosensitization to only one fungal allergen was observed in 14.5% of the patients. *A. alternata*, *A. fumigatus*, and *C. albicans* were the most prevalent monosensitizers (37.9%, 27.6%, and 20.7%, respectively).

In agreement with Amado et al. [45], who reported that patients sensitized to *Alternaria* spp. are probably sensitized to one or more other allergenic fungal species, our result revealed that sensitization to *A. alternata* was rampant in 86.2% of *A. niger*, 73.4% of *A. fumigatus*, 77% of *P. notatum*, and 56.5% of *C. albicans* sensitized patients. Although cross-reactivity among fungi is not fully understood, multiple fungal sensitizations are due to the sensitivity to multiple antigens, whether cross-reactive or not [49]. Moreover, the combined fungal sensitization pattern observed in studied patients supports the imposition of mixed mold extract in SPT during screening for implicate fungal aeroallergens to circumvent the vast number of allergenic fungal species, which impedes the significant progress in fungal extract standardization [26].

Only two patients lacked specific IgE immunoreactivity to any fungal allergens on an immunoblot when mixed mold-sensitized patients were categorized according to their SPT reactivity patterns. This finding may be attributed to the inclusion of *Cladosporium* allergen in the mixed mold extract used in SPT and its absence in the immunoblot assay panel. Conversely, six patients displayed specific IgE immunoreactivity on immunoblots despite their negative SPT results. Further analysis could identify these six patients as *C. albicans* monosensitizers. However, this finding can be explained in a binary manner: the studied patients are either sensitized to *C. albicans*, which is not supported by the limitation of relevant SPT or sensitized to a fungal species cross-reactive with *C. albicans* [50].

A significant finding of the current study was the increased prevalence of *C. albicans* sensitization in adults, despite the inability to detect a correlation between sociodemographic and clinical characteristics and fungal sensitization patterns. However, the increased prevalence of *C. albicans* sensitization was previously reported by Ezeamuzie et al. [50], Fernández-Soto et al. [18], Asero and Bottazzi [51], and Mari et al. [12]. *C. albicans* can be found in many niches in the body, including the digestive tract and vagina, as a commensal organism and less commonly as an opportunistic pathogen, albeit its role in airway allergy is debated [52,53]. The increased risk of local infections may reflect the correlation between *C. albicans* sensitization and allergic rhinitis alone or in combination with asthma. Improper administration of antibiotics may result in commensal *Candida* overgrowth above the tolerance threshold, and hence, sensitization proceeds [50,54]. In context, the significant association between allergic conjunctivitis with allergic rhinitis alone or combined with asthma may validate the significant correlation of *C. albicans* sensitization to allergic conjunctivitis. Therefore, these data should encourage further investigations on the relationship between exposure to *Candida* infections and the development of airway-allergic disease [55].

Although approximately two-thirds of patients sensitized to molds were residents in rural areas, there was no association between the residence of the studied patients and fungal sensitization patterns, which aligns with the findings of previous studies conducted in India [56] and the USA [57]. This finding could be related to the nature of the Zagazig locality, the origin of the inhabitants, and the air concentration of fungal spores. Zagazig is the capital city of Al-Sharqia province, which is an agricultural province in nature and the third-largest Egyptian province in terms of population. The Zagazig center consists of Zagazig city, surrounded by 75 villages [58], with almost all its urban citizens having rural origins [59]. Moreover, AIU, Faculty of Medicine, Zagazig University is the only specialized place for allergy testing and immunotherapy for residents in the locality. In addition, it was estimated that the rural environment in Egypt had a higher concentration of fungal spores than the urban one [7]. 

As previously reported, up to 40% of allergic rhinitis patients have combined allergic rhinitis and asthma [60], albeit patients with combined allergic rhinitis and asthma represented 30.9% of allergic rhinitis patients in the current study. In addition, approximately fifty percent of patients with upper or lower airway allergy demonstrated fungal sensitization, which was not limited to allergic rhinitis alone. It is noteworthy that exposure to fungal aeroallergens may exacerbate allergic rhinitis and asthma to respiratory symptoms aggravation [60], particularly in asthmatic children [61]. 

Our results revealed that the total IgE serum level did not correlate with sociodemographic or clinical characteristics of the studied patients nor differed between fungal sensitization patterns. Despite early studies [62,63] considering total IgE serum level the most straightforward tool to recognize atopic patients, low or normal values cannot exclude IgE-mediated allergy. Conversely, values higher than usual are associated with atopic patients and parasitic infestations and malignancies [64]. In addition, total IgE serum level correlates to the severity of allergic diseases [64,65] rather than sensitization patterns. Therefore, the absence of significant total IgE serum level variations between mixed mold-sensitized and non-sensitized patients was not unexpected, since all the studied patients in the current study were experiencing allergic airway diseases.

In order to gain a better understanding of the mechanisms and pathways by which fungi trigger allergic airway diseases and exacerbate asthma, additional research is recommended. It was reported that frequent exposures to fungal allergens induce T helper 1, 2, and 17 immune responses and chronic inflammation in airways [66]. Conserved fungal cell walls components such as β-glucans, chitin, and proteases are targeted and recognized by the immune system [67]. In response to these components, epithelial cells secrete cytokines, chemokines, and antimicrobial peptides. In addition, mice models demonstrated that innate lymphoid 2 cells contribute to the triggering and persistence of fungus-mediated allergic responses via the release of proinflammatory cytokines such as IL-5 and IL-13. Nevertheless, the mechanisms by which proinflammatory cytokines such as IL-33 activate innate lymphoid cells, eosinophils, and macrophages involved in the pathogenesis of fungal allergy need further investigation [68].

Eventually, identifying fungal aeroallergen sensitization patterns in airway-allergic patients is essential to avoid environmental trigger exposure and reduce allergy symptoms. It paves the way for patient education and tailoring personalized treatment with specific allergen immunotherapy [69]. 

## 5. Conclusions

Sensitization to fungi is the fourth prevalent aeroallergen implicated in airway-allergic diseases in the Zagazig locality. Monosensitization to fungal aeroallergen is remarkably infrequent, whereas sensitization to *A. alternata* has the highest prevalence rate among allergenic fungi in airway-allergic patients. Patients sensitized to one fungal species tend to be sensitized to one or more other species suggesting cross-reactivity between different allergenic fungi. Fungal sensitization patterns can be screened for SPT and confirmed by fungal-specific IgE immunoassay. Identifying fungal aeroallergen sensitization patterns in the study locality will improve airway allergic patients’ management. In light of the environmental data of airborne fungal spores, it is essential for individualized allergen avoidance and patient education, tailoring a better immunotherapeutic approach and encouraging government control efforts.

## Figures and Tables

**Figure 1 jof-09-00185-f001:**
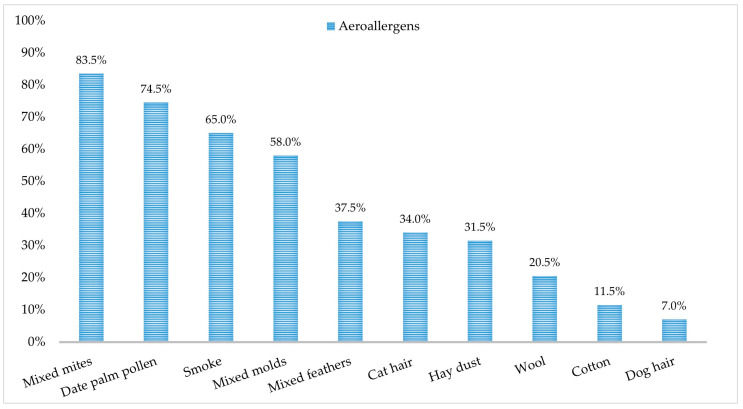
Aeroallergens sensitization patterns by skin prick test in the studied patients (N = 200).

**Figure 2 jof-09-00185-f002:**
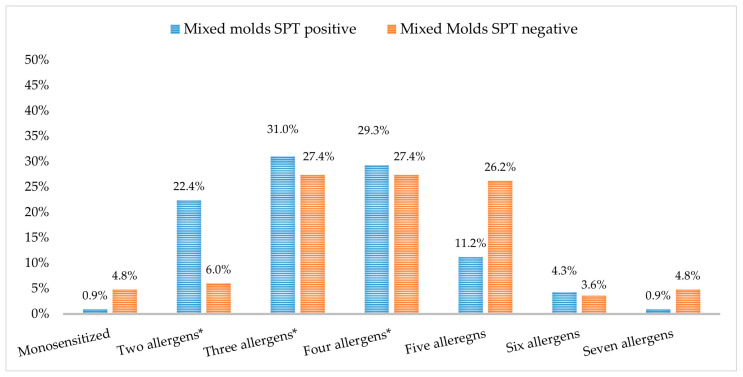
Polysensitization patterns in the studied patients by skin prick test (SPT). Data were analyzed by Fisher exact test, and significant differences were defined as * *p* = 0.001 (N = 200).

**Figure 3 jof-09-00185-f003:**
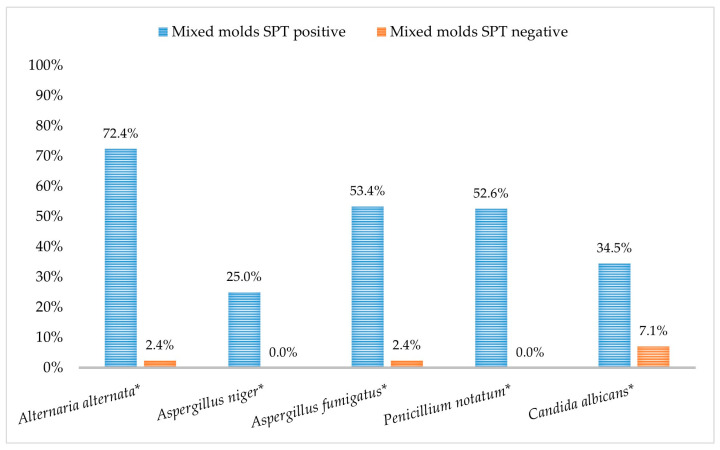
Fungi sensitization patterns in the studied patients by immunoblot assay. Data were analyzed by Fisher’s exact test, and significant differences was defined as * *p* < 0.001 (N = 200).

**Figure 4 jof-09-00185-f004:**
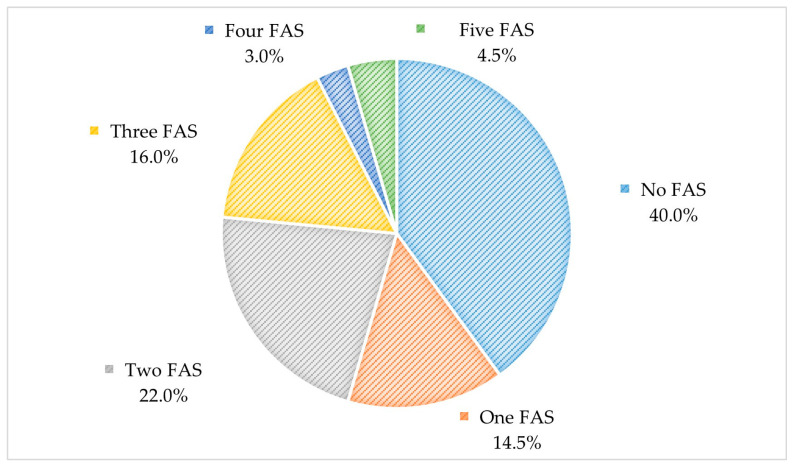
Polysensitization patterns to fungal aeroallergens in the studied patients by immunoblot assay (N = 200). FAS, fungal aeroallergen sensitization.

**Figure 5 jof-09-00185-f005:**
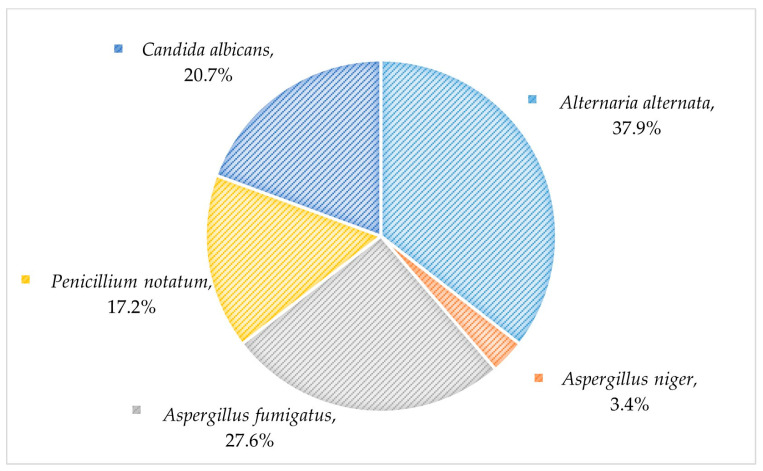
Fungal aeroallergen monosensitization rate in the studied patients by immunoblot assay (n = 29).

**Figure 6 jof-09-00185-f006:**
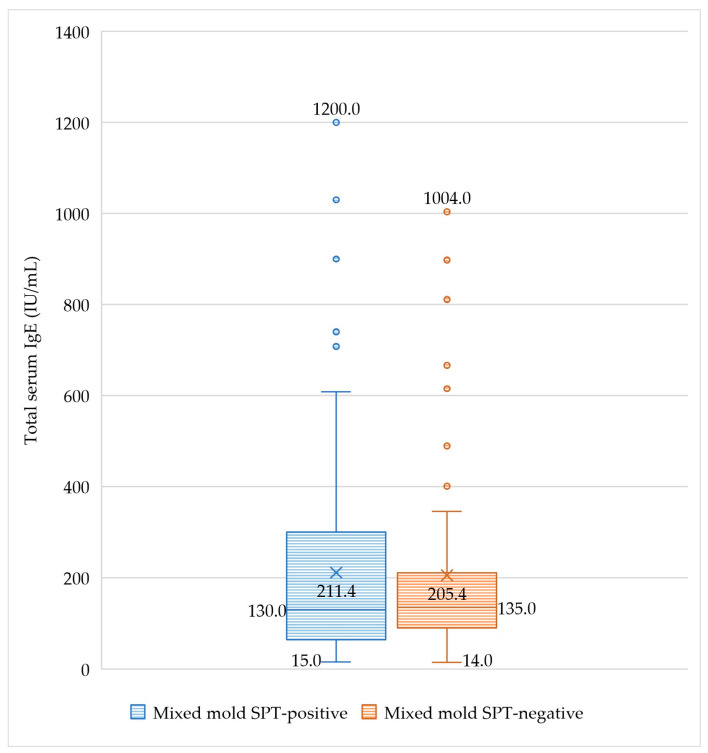
Total IgE serum level (median and range) difference between mixed mold skin prick test (SPT)-positive and negative patients. Data were analyzed by Mann–Whitney U test (N = 200). ×, mean.

**Figure 7 jof-09-00185-f007:**
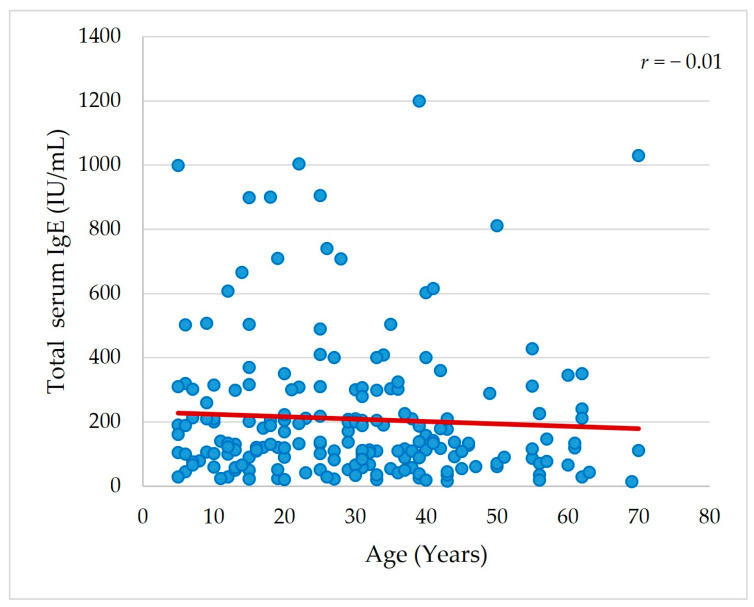
Correlation between age (*X* axis) and total IgE serum level (IU/mL) (*Y* axis) in the studied patients. Correlation was measured by Spearman’s rank correlation coefficient (*r*) (N = 200).

**Table 1 jof-09-00185-t001:** Sociodemographic and clinical characteristics of the studied patients.

Variable	(N = 200)	%
**Age** (Years)		
Mean ± SD	29.2 ± 15.8	
Median (Min–Max)	29 (5–70)	
Children (1–17 y)	54	27
Adults (≥18)	146	73
**Sex**		
Female	79	39.5
Male	121	60.5
**Residence**		
Urban	64	32.0
Rural	136	58.0
**Type of airway allergy**		
Allergic rhinitis	121	60.5
Allergic asthma	25	12.5
Combined allergic rhinitis and asthma	54	27.0
**Associated allergic conditions**		
Urticaria ^a^	9	4.5
Allergic conjunctivitis ^b^	23	11.5

^a^ Non-significant correlation to type of airway allergy (*p* = 0.09). ^b^ Significant correlation to type of airway allergy (*p* = 0.038).

**Table 2 jof-09-00185-t002:** Variables associated with patterns of mixed molds sensitization in studied patients (N = 200).

Variable	Mixed Mold SPT-Positive n = 116 (%)	Mixed Mold SPT-Negative n = 84 (%)	Test of Significance	*p*-Value
**Age** (Years)			Mann–Whitney U	0.9
Mean ± SD	29.5 ± 15.5	30.2 ± 17.0		
Children (1–17) (n = 54)	30 (25.9)	24 (28.6)	χ2 test	0.7
Adults (≥18) (n = 146)	86 (74.1)	60 (71.4)		
**Sex**			χ2 test	0.5
Female (n = 79)	48 (41.4)	31 (36.9)		
Male (n = 121)	68 (58.6)	53 (63.1)		
**Residence**			χ2 test	0.7
Urban (n = 64)	36 (31.0)	28 (33.3)		
Rural (n = 136)	80 (68.0)	56 (66.7)		
**Type of airway allergy**			Fisher exact	0.6
Allergic rhinitis (n = 121)	71 (61.2)	50 (59.5)		
Allergic asthma (n = 25)	12 (10.3)	13 (15.5)		
Combined (n = 54)	33 (28.5)	21 (25.0)		
**Associated allergic conditions**				
Urticaria (n = 9)	4 (3.0)	5 (6.0)	Fisher exact	0.5
Allergic conjunctivitis (n = 23)	16 (14.0)	7 (8.3)	χ2 test	0.2
**Specific IgE Immunoblot assay ^a^**				
Fungal allergen sensitization (n = 120)	114 (98.3)	6 (4.8)	Fisher exact	<0.001 *

Significant difference was defined as * *p* ≤ 0.001. ^a^ Dependent variable.

**Table 3 jof-09-00185-t003:** Variables associated with patterns of fungal aeroallergen sensitization in studied patients (N = 200).

Variable	*A. alternata* n = 86 (43.0%)	*A. niger* n = 29 (14.5%)	*A. fumigatus* n = 64 (32.0%)	*P. notatum* n = 61 (30.5%)	*C. albicans* n = 46 (23.0%)
**Age** (Years)					
Children (1–17) (n = 54)	18 (20.9)	5 (17.2)	14 (21.9)	15 (24.6)	7 (15.2)
Adults (≥18) (n = 146)	68 (79.1)	24 (82.8)	50 (78.1)	46 (75.4)	39 (84.8)
*p*-value	0.09 ^a^	0.2 ^a^	0.3 ^a^	0.6 ^a^	0.04 *^,a^
**Sex**					
Female (n = 79)	39 (45.3)	11 (37.9)	31 (48.4)	22 (36.1)	16 (34.8)
Male (n = 121)	47 (54.7)	18 (62.1)	33 (51.6)	39 (63.9)	30 (65.2)
*p*-value	0.1 ^a^	0.9 ^a^	0.08 ^a^	0.5 ^a^	0.5 ^a^
**Residence**					
Urban (n = 64)	29 (33.7)	12 (41.4)	21 (32.8)	20 (32.8)	15 (32.6)
Rural (n = 136)	57 (66.3)	17 (58.6)	43 (67.2)	41 (67.2)	31 (67.4)
*p*-value	0.7 ^a^	0.2 ^a^	0.9 ^a^	0.9 ^a^	0.9 ^a^
**Type of airway allergy**					
Allergic rhinitis (n = 121)	50 (58.1)	18 (62.1)	45 (70.3)	39 (63.9)	36 (78.3)
Allergic asthma (n = 25)	11 (12.8)	2 (6.9)	5 (7.8)	4 (6.6)	3 (6.5)
Combined (n = 54)	25 (29.1)	9 (31.0)	14 (21.9)	18 (29.5)	7 (15.2)
*p*-value	0.8 ^a^	0.6 ^b^	0.1 ^a^	0.2 ^b^	0.02 *^,b^
**Associated allergic conditions**					
Urticaria (n = 9)	2 (2.3)	1 (3.4)	1 (1.6)	3 (4.9)	3 (6.5)
*p*-value	0.2 ^b^	0.6 ^b^	0.2 ^b^	0.6 ^b^	0.3 ^b^
Allergic conjunctivitis (n = 23)	11 (12.8)	3 (10.3)	10 (15.6)	8 (13.1)	9 (19.6)
*p*-value	0.6 ^a^	0.6 ^b^	0.2 ^a^	0.6 ^a^	0.05 *^,a^
** *A. alternata* ^c^ **					
Sensitized (n = 86)	85 (100.0)	25 (86.2)	47 (73.4)	47 (77.0)	26 (56.5)
*p*-value	1	<0.001 **	<0.001 **	<0.001 **	0.035 *
**Mixed mold SPT ^c^**					
Positive (n = 116)	84 (97.7)	29 (100.0)	62 (96.9)	61 (100.0)	40 (87.0)
*p*-value	<0.001 **^,b^	<0.001 **^,b^	<0.001 **^,b^	<0.001 **^,b^	<0.001 **^,a^

Significant difference was defined as * *p* ≤ 0.05, ** *p* ≤ 0.001. ^a^ Data were analyzed by Pearson Chi-Square test. ^b^ Data were analyzed by Fisher’s exact test. ^c^ Independent variable.

## Data Availability

The data of this study are available from the corresponding author on reasonable request.

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
