# Peer review of "Fungal Aeroallergen Sensitization Patterns among Airway-Allergic Patients in Zagazig, Egypt"

_jof, 2023, doi:10.3390/jof9020185_

Round 1

Reviewer 1 Report

In this manuscript, the authors investigated fungal aeroallergen sensitization patterns among airway-allergic patients in Zagazig, Egypt. They showed fungal sensitization is the fourth leading aeroallergen in that area and Alternaria alternata is the highest prevail fungi among different species. In addition, they showed patients sensitized to one fugal species tend to be sensitized to one or more other species instead of mono-sensitization. Although they nicely showed these results with SPT screening and fungal specific IgE titers, it would increase the quality of the manuscript if they add more immunological data such as different fugal associated antigen stimulation of T cells and the tendency of T helper type 2 associated responses by analyzing cytokines, chemokines, and other associated factors. Lastly, some of typo errors should be corrected. 

Reviewer 2 Report

The paper “Fungal Aeroallergen Sensitization Patterns Among Airway-Allergic Patients in Zagazig, Egypt” is a great contribution related to population-response to fungal and other aeroallergens present at that region. The specific aeroallergen-sensitization test performed is also a contribution to the health sector and academy.

Some questions that the authors can consider in the manuscript are:

Abstract:

·         Correct in abstract lines 29-30 “The present work aimed to identify the common fungal aeroallergens prevalent in the Zagazig locality, Egypt” , since the research focus on  identify population sensitization to different aeroallergens.

·         It’s mandatory mention in the abstract the main of this research, his impact and its application.

Introduction:

·         Line 51, change “climatic circumstances” by weather conditions

·         The problem about fungal spores in air related to air quality and health impact must be included.

·        A background of fungal spores in air at the study region must be included

·        In the final part of the introduction the contribution, impact and application of this research must be included.

Materials and Methods:

·         This section must include the environmental settings of the study region, moreover related to weather conditions.

·         Lines 98-99, explain why authors selected only those species of fungi? Why only four species? Why did not included by example Cladosporium a frequent fungal spore in air.

·         Lines 123-130, The statistical analysis must specify which variables were taken into account: dependent and independent in all the test performed.

Results:

·         Line 175 correct the subtitle say “scoiodemograohic” must say sociodemographic.

·         Check spelling and language editing at this section.

Discussion:

·         Lines 216-221 are not discussion; it must be moved to conclusion section or remove.

·         line 269 scientific names must be in italics.

·         Lines 307-319 the authors should explain why are more positive-tested patients at rural than urban areas?

·         Authors should explain why there was no significant variation in total IgE serum level between sensitized and non-sensitized patients to mixed molds aeroallergen?

·         A discussion section about the prevalence of fungal spores in air in the study region is lacking. It’s important to show data of which fungal spores are prevalent in air to compare with the results of specific fungal species-test.

Conclusion:

·         The contribution and relevance of this results are lacking at the conclusion, explain the application of the results in the health and governmental sectors.

Figures and Tables

Figure 2: reduce the numbers in the graph (percentages) to avoid overlapping

Figure 3: put the scientific names in italics in the x-axis of the graph

Figure 6: reduce the numbers inside the graph to better show them.

Round 2

Reviewer 2 Report

I appreciate that authors included all my recommendations in this new version of the manuscript.